# Resistance of Common Bean Genotypes to *Chrysodeixis includens* (Walker, 1858) (Lepidoptera: Noctuidae)

**DOI:** 10.3390/insects14120905

**Published:** 2023-11-23

**Authors:** Marcelo Augusto Pastório, Vanda Pietrowski, Adriano Thibes Hoshino, Luciano Mendes de Oliveira, Fernando Teruhiko Hata, Maurício Ursi Ventura, Humberto Godoy Androcioli

**Affiliations:** 1Departamento de Agronomia, Universidade Estadual do Oeste do Paraná, Rua Pernambuco, 1777-Centro, Marechal Cândido Rondon 85960-000, Paraná State, Brazil; marcelo.pastorio@hotmail.com (M.A.P.); vandapietrowski@gmail.com (V.P.); 2Departamento de Agronomia, Universidade Estadual de Londrina, Rodovia Celso Garcia Cid, PR-445, Km 380-Campus Universitário, Londrina 86057-970, Paraná State, Brazil; hoshinoagro@gmail.com (A.T.H.); luciano.agro.oliveira@gmail.com (L.M.d.O.); mventura@uel.br (M.U.V.); 3Departamento de Agronomia, Universidade Estadual de Maringá, Av. Colombo, 5790-Jd. Universitário, Maringá 87020-900, Paraná State, Brazil; hata.ft@hotmail.com; 4Laboratório de Entomologia, Instituto de Desenvolvimento Rural do Paraná IAPAR-EMATER, Rodovia Celso Garcia Cid, km 375—Conjunto Ernani Moura Lima II, Londrina 86047-902, Paraná State, Brazil

**Keywords:** antibiosis, antixenosis, secondary compounds, varietal resistance, plant breeding

## Abstract

**Simple Summary:**

The common bean is an important low-cost protein source in developing countries worldwide. The soybean looper causes defoliation impairing common bean production. The objective was to study the resistance of common bean genotypes toward soybean looper. The biology, fertility life table, oviposition preference and free-choice feeding preference were analyzed. Phenol and flavonoid content were assessed in the leaves of the evaluated genotypes. Uirapuru genotype negatively affected the soybean looper biology and reproduction. Tangará genotype favored these parameters. Genotypes Quero-Quero, Nhambu, Corujinha, Andorinha, ANFC 9, Siriri, BRS Radiante and Verdão were more attractive for third-instar caterpillars. Common bean genotypes with higher contents of phenolic and flavonoid compounds decreases the soybean looper survival rate. Common bean genotypes with dark leaves are less preferred for oviposition.

**Abstract:**

The common bean (*Phaseolus vulgaris* L.) is an important leguminous crop providing low-cost protein in developing countries worldwide. Insect pests are the main threats to common bean production, and this article focuses on the soybean looper (SL) *Chrysodeixis includens* (Walker, 1858) (Lepidoptera: Noctuidae), which feeds on leaves and pods. The recurrent use of synthetic chemicals may lead to pest resistance. Genetically resistant plants may diminish their use. Thus, the objective was to study common bean genotypes’ resistance toward SL. The plants were grown in greenhouse conditions. The biology, fertility life table, oviposition preference and free-choice feeding preference were analyzed. Phenol and flavonoid content were assessed in leaves using a biology assay. Uirapuru genotype negatively affected *C. includens* biology and reproduction. Tangará genotype favored these parameters. Genotypes Quero-Quero, Nhambu, Corujinha, Andorinha, ANFC 9, Siriri, BRS Radiante and Verdão were more attractive for third-instar larvae. Negative correlations between phenolic and flavonoid compounds with survival rate (from egg to adult) rate were found. Common bean genotypes with dark leaves are less preferred for oviposition.

## 1. Introduction

The common bean, *Phaseolus vulgaris* L., is an important source of protein, vitamins, and minerals [1] for more than 300 million people worldwide [2]. In general, this crop constitutes an income source for small-scale farmers [3].

Brazil is the world’s third-largest producer and consumer, with an average yield of 1000 kg ha^−1^, encompassing three million hectares [4]. In Paraná State are cultivated 383 thousand hectares through in three yearly cultivation seasons [5]. However, both biotic and abiotic factors may diminish yields. Different arthropod pests may damage plants at all plant development stages [6,7].

The soybean looper (SL), *Chrysodeixis includens* (Walker, 1858) (Lepidoptera: Noctuidae), has become increasingly important and is commonly found in annual leguminous crops, most importantly in soybean and common-bean cultivation areas, due to the high defoliation potential, which can reach 80 percent leaf consumption in some areas, resulting in yield losses and increasing cultivation costs due to control necessity [8,9,10]. After soybean looper foliar feeding, only the veins remain intact, negatively affecting the plant’s photosynthetic area, resulting in disease gateways [11,12].

To date, the main control method is synthetic insecticide spraying. However, excessive use, disregarding technical criteria, has resulted in the selection of resistant populations and secondary infestations [13,14]. The development of complementary methods and management strategies, including insect-resistant plant breeding, is necessary [15,16].

Plant resistance may be effective in reducing insect pest populations, keeping populations below the economic damage threshold [17]. In addition, plant resistance is generally compatible with other control measures, which is ideal for integrated pest management (IPM) programs due to the adoption feasibility. Forms of resistance include antixenosis, antibiosis and tolerance [18]. Understanding these types of resistance may provide suitable information for the selection of resistant genotypes [16,19,20]. 

A crucial factor relating to plant–insect antixenosis is the insect’s perception during host selection, which is determined by various factors, such as the leaf’s coloration and reflectance [21,22,23]. The study of leaf coloration parameters may prove useful in insect-resistant plant breeding programs as they could influence the herbivorous insect’s behavior.

When insects feed on plants that express antibiosis, their biology can be affected, resulting in reductions in size, weight, metabolic processes and even higher mortality in immature stages (larvae and pupae). Antixenosis characteristics can influence insect behavior when choosing a host and can also alter insect development, leading to negative effects that may be like those caused by antibiosis, making it difficult to distinguish. Plants that expressing tolerance not affect the insect behavior nor impaired its development, these plants are lesser damaged than other genotypes under the same infestation conditions [17,18,24]. The scarcity of studies on the resistance mechanisms present in common bean genotypes are the main impediments toward obtaining insect-resistant genotype [25]. Considering the SL’s high potential to damage common bean, twenty-two common bean genotypes were chosen to assess possible resistance mechanisms and potential genotypes as sources of resistance for breeding programs.

## 2. Materials and Methods

Bioassays were conducted under controlled conditions (T 25 ± 3 °C, 70 ± 10% RU and photophase 14:10 L:D). The selection of common bean (*Phaseolus vulgaris* L.) genotypes was based on market availability and their differing phenotype characteristics. The common bean genotypes were divided into two randomly assigned groups of eleven to facilitate assay handling (Table 1). Experiments were performed to assess soybean looper (*Chrysodeixis includens* Walker) (SL) larvae feeding preference, leaf consumption and oviposition preference (adults). The IPR Uirapuru genotype served as susceptibility standard due to results previously reported in field crop and V.C.U. (values of cultivation and use), this genotype was assigned as the twelfth genotype of each group. The IPR Uirapuru genotype is an undetermined type II development black-type bean. All seeds were provided by the Instituto de Desenvolvimento Rural do Paraná IAPAR-EMATER (IDR—Paraná), Londrina, Paraná State, Brazil.

### 2.1. Insect Rearing 

The insect eggs were obtained from the IDR—Paraná’s entomological rearing laboratory. Insects were then reared in plastic cups (50 mL) filled with 10 mL of artificial diet [26] under controlled conditions (T 25 ± 3 °C, UR 70 ± 10%, 14:10 L:D). Pupae were sexed and grouped in 20 couples, in plastic cages (21 cm high × 15 cm diameter) lined with paper filter for adult emergency and mating. A solution containing distilled water, methylparahydroxybenzoate (1%) and honey (6%) was soaked in cotton for sustenance. 

Bond A4 paper sheets were placed in the cages as egg substrate. These were replaced every other day, and the removed substrate paper was cut in pieces, with each piece containing fifty eggs, that were placed in 500 mL containers for egg hatching. 

### 2.2. Plant Cultivation 

All genotypes were cultivated in 4 L pots filled with Eutropherric red latosol (oxisols) and irrigated two times each day without prior fertilizing. The plants were kept inside a greenhouse in IDR—Paraná (23°21′20.6″ S and 51°09′57.3″ W).

### 2.3. Bioassays

#### 2.3.1. Multiple-Choice Feeding Preference Assay

In this bioassay, a randomized block design was adopted with two groups of 12 genotypes with 15 replications. Each replicate consisted of a cylindrical container (25 cm in diameter × 5 cm in height) containing filter paper moistened at the base, in which 12 leaf discs with 3.14 cm^2^ (one of each genotype, including IPR Uirapuru) were deposited in a circular and equidistant arrangement close to the edge of the container. At each repetition, the order of the leaf discs in the circular arrangement was randomized.

Leaf discs were obtained from plants with 25 days of emergence. The leaves were collected and stored in plastic bags, placed in a cooler containing ice to preserve leaf turgency, they then cut into discs with a sterilized metal circular cutting tool (2 cm in diameter) before infestation of the larvae. 

In the center of the container, 36 third-instar larvae were placed after a 12-h fasting. The number of larvae present in each leaf disc was quantified 12 h after infestation. This design, which contrasted 12 genotypes at once, was used to mitigate time constraints, given the amount of combination required for a contrast with a smaller number of genotypes at a time. The collected data was used to calculate the attractiveness index (AI), using the formula:AI = 2T/(T + P)
where T = the number of insects attracted to the genotype, and P = the number of insects attracted to the susceptibility standard genotype. 

The AI index varies from 0 to 2: values lower than 1 indicate less attractivity compared to the control; 1 indicates similar attractivity; and 2, higher attractivity [27].

#### 2.3.2. Foliar Feeding Index 

Like the method described above, this assay followed a randomized block design with two groups of 12 genotypes and 15 replications. The same container and arrangement of leaf discs were used. However, only 12 third-instar larvae were released in the center of the container (one per foliar disk). After 24 h the leaf disks leftovers were recovered, placed in paper bags, and dried in a forced air oven at 60 °C [28]. Control disks from each genotype were also collected and dried to determine the standard weight. The consumption recorded was transformed into the correspondent area (cm^2^). The feeding index (FI) was determined using the following formula [29]:FI = [TD × FAd] − [(FAd × DW)/DWd]
where TD—total disks, FAd—foliar area of one disk, DWd—dried weight of one disk, and DW—leftover dried weight. 

#### 2.3.3. Oviposition Preference

The fifteen genotypes with the lowest feeding index (FI) were then selected for oviposition preference assay: Campos Gerais, BRS Esteio, Uirapuru, Sabiá, ANFC 9, Juriti, Eldorado, Quero-Quero, Verdão, Curió, Gralha, Tangará, Tuiuiú, IAPAR 81 and Capitão. 

The genotypes were cultivated under the same conditions previously described. Pots containing plants with the third trifoliolate leaves fully expanded were placed in a circular and equidistant arrangement close to the edge of the voile cages (2 m × 2 m × 2 m). Thirty SL moths couples were released in the center of the cages, after 72 h the number of eggs deposited in each plant was quantified. The oviposition preference index (OPI) was calculated by using the formula [30,31]:$$OPI=\frac{T-P}{T+P}\times 100$$
where T = the number of eggs recorded in the genotype, and P = the number of eggs in the standard genotype. 

Standard deviation (SD) was calculated to establish the superior and inferior limits of the comparisons. Genotypes with OPI values higher than the superior limits were considered as attractants, and those lower than the inferior ones as deterrents.

### 2.4. Leaves Coloration 

Leaf coloration was evaluated as a probable reason for SL deterrence or attractivity. Five replications of the same 15 genotypes used in the oviposition preference assay were sowed in propylene pots (4 L) filled with Eutrophic Red Oxisol. The color attributes were measured when the plants were with the third trifoliolate leaves fully expanded, five measurements per replicates were done. The light intensity (L*), hue (a*) and saturation (b*) of leaves of each genotype was determined using a colorimeter (Konica Minolta Business Solutions do Brasil Ltd.a.^®^, Florianópolis, Santa Catarina State, Brazil, Chroma Meter CR–400, color system CIE (Commission Internationale de L’Éclairage)) [32]. The leave’s adaxial central portion in the plant’s upper third region was used. A black background was used behind the leaf to avoid reflected sunlight influence [33].

### 2.5. Biology and Reproductive Parameters

The genotypes Campos Gerais, BRS Esteio, ANFC 9, Juriti, Quero-Quero, Eldorado, Capitão, Curió, Verdão, Tangará, Gralha, IAPAR 81 and Tuiuiú were selected due to their low AI values.

Seventy neonate larvae were placed in plates (Petri dishes) lined with filter paper moistened with distilled water. The plates were clustered in seven groups of 10 plates each. The larvae were reared from leaves of the plant’s upper third region, from the selected genotypes, which were in the blossom development stage. The following variables were evaluated: (i) duration of each instar and total larval period; (ii) larval survival rate; (iii) prepupal period; (iv) prepupal survival rate; (v) pupal duration period; (vi) pupae weight (96 h after pupation); (vii) pupal survival rate and deformations; (viii) sexual ratio [34]; (ix) pre-oviposition and oviposition; (x) daily and total fecundity; (xi) fertility and (xii) period of eggs incubation. 

### 2.6. Fertility Table 

The life and fertility tables were elaborated from biology and reproductive parameters results of SL, for each common bean genotype the following parameters were estimated: age interval (x), specific fertility (mx), survival rate (lx), net reproduction rate (R0), intrinsic growing rate (rm), mean generation time (T), double population time (Td) and finite rate of population increase (λ) (Jackknife method). Life Table package from SAS software version 9.4 was used according to previous study [35]. 

### 2.7. Total Phenols and Flavonoids 

The same genotypes used in the biology and reproductive parameters assay were evaluated for phenol compound and flavonoid s by collecting one leaf of the median third. An aliquot of 1 g fresh weight was prepared by adding 10 mL of ethanol 70% (*v*/*v*) suspended and shaken for 30 min (Nova Ética Produtos e Equipamentos Cientificos, Ltd.a., Vargem Grande do Sul, São Paulo State, Brazil, agitator model 109) in ambient conditions. Next, the extract was centrifuged at 1013× *g* (Fanem, Ltd.a., Guarulhos, São Paulo State, Brazil, Excelsa 2 model 205N) for five minutes, and the resulting supernatant was used for analysis.

For phenolic levels, 1.0 mL of the ethanolic extracted was mixed with 1.0 mL of methanol, 1.0 mL of Folin–Ciocalteau reagent 0.2 N and 1.0 sodium carbonate 10% (*m*/*v*). The obtained mixture was placed in a dark environment for 30 min at 25 °C. Absorbance was then measured at 765 nm wavelength in the spectrophotometer (Micronal S.A., São Paulo, São Paulo State, Brazil, model AJX1600). Gallic acid was used as standard in the concentrations of 10.0–100.0 mg L^−1^. Results were expressed in mg equivalent of gallic acid per 100 g of sample (mg GAE 100 g^−1^) [36].

Total flavonoid was analyzed using 1.0 mL of the methanolic extract, aluminum chloride at 5.0% (*m*/*v*) and 2.0 mL of methanol, kept in a dark environment for 30 min. Samples were read in the spectrophotometer (Micronal S.A., São Paulo, São Paulo State, Brazil, model AJX1600) at 425 nm. Quercetin was used as a standard with concentrations of 50.0–500.0 mg L^−1^, and the results were expressed in quercetin equivalent (QE), in milligrams per 100 g sample [37].

### 2.8. Statistics 

The assumptions of parametric analysis of variances, homogeneity and normality of errors, were assessed (Hartley’s Fmax and Shapiro–Wilk tests, respectively). When assumptions were met Scott–Knott test (*p* < 0.05) was used to compare means. Otherwise, the Kruskal–Wallis test was applied, and the means were compared using Student–Neuman–Keuls (SNK) by using the SISVAR software [38]. For the variables, deformed pupae and adults, sexual ratio and eggs survival rate, the chi-square test was used. Pearson correlation was also applied to verify the association between the biologic parameters vs. chemical compounds of *C. includens* determined in the genotypes. 

## 3. Results

### 3.1. Feeding Preference Assays 

The attractivity index for soybean looper (SL) third-instar larvae was used to form two groups between the genotypes, as follows: For group 1, the genotypes MD 1133, Campos Gerais, Garça, Celeiro and Curió were treated as repellent in comparison to Uirapuru (Figure 1A), and the remaining were classified as attractive. For the group 2, Quero-Quero and Nhambu were classified as attractive compared to Uirapuru, and the remaining genotypes as repellent (Figure 1B).

#### Foliar Feeding 

The ANFC 9 genotype was the most consumed from group 1 (Table 2). Celeiro, MD 1133 and Siriri were also more consumed than the standard genotype Uirapuru. The genotypes Curujinha, Garça and Curió were similar, and the remaining materials were the least consumed. From the second group, Gralha and IAPAR 81 were consumed more than the remaining genotypes, including the standard Uirapuru.

### 3.2. Free-Choice Oviposition Preference 

No significant difference was found regarding SL oviposition preference, the number of eggs per plant varied from 80.8 to 146.2 (Table 3). Previous studies also failed to obtain differences among common beans genotypes regarding oviposition preference (151.0 to 234.8 eggs per plant) [39].

Considering the OPI, the genotypes Verdão, Tuiuiú, Gralha, Eldorado, Sabiá and BRS Esteio were classified as oviposition deterrents when compared to Uirapuru due to calculated values exceeding the inferior threshold of the standard deviation of the sample (SD ± 8.64) (Figure 2). The remaining genotypes were neutral compared to the standard due to OPI values fitting inside the SD interval (+8.64 and −8.64) (Figure 2).

### 3.3. Leaves Color 

For the genotypes Eldorado, Uirapuru, BRS Esteio, Capitão and Gralha, the L* values were higher than the remainders, evidencing higher luminosity. In opposition, lower values were obtained for ANFC 9, Juriti, Sabiá, Tuiuiu, Curió, Iapar 81, Campos Gerais and Verdão, characterized by less luminosity or dark leaves (Table 4).

When far from the host plant, luminosity and object size contrasting the horizon, constitute the first stimulus when pests choose the host plant, as they have no or little plant-hue perception. However, when close to the host plant, spectral quality, particularly hue and intensity, appears predominant on host plant detection [40]. The a* values (hue) are negative, evidencing green coloration, which is far from a red color. Variations were found among genotypes, establishing two distinct groups: Eldordo, Uirapuru, Tangará, BRS Esteio and Capitão which have values below −17; and another group, including all remaining materials, which has values exceeding −16 (Table 4). In this case, the lower values demonstrate the intensification of the green color levels in the leaves. 

The b* values (saturation) indicate the blue color (negative values) and the yellow color (positive values). Variations were observed among genotypes, dividing them into three different groups. The positive values found for all the genotypes screened indicate the predominance of the yellow color compared to blue. The first group was composed by genotypes Eldorado, Uirapuru and BRS Esteio, for which the values were above 22; for the intermediate group, including Tangará and Capitão, the values were close to 20; and for the third group, including the remaining genotypes, the values were inferior to 19 (Table 4).

The configuration, including a* negative and b* positive (in higher degrees) (green and yellow pigments, respectively), may constitute the visual stimulus of an ordinary leaf, leading to positive visual responses of herbivorous insects. Plant spectral quality, mostly hue and intensity, is probably the main reason for the insect’s choice [41,42,43].

A significant negative (−0.62) Pearson correlation (*p* = 0.012) between common bean’s leaves luminosity (L*) and the OPI was obtained, suggesting that, darker leaves negatively affect SL oviposition, corroborating previous studies with similar studying parameters [39]. 

### 3.4. Biology and Reproductive Parameters 

Differences in incubation period were established among genotypes from 3.2 (Campos Gerais) to 3.7 days (IAPAR 81 and Uirapuru). (Table 5). The egg survival rate also was influenced by the genotypes, from 25.41 (Gralha) to 59.19 (Quero-quero) (Table 6). In previous studies, the egg survival rate reached 81.7 [36] and 85% when insects were fed common bean genotypes or an artificial diet [44].

The duration of the larval period varied from 14.5 to 16.6. These values were close to those obtained previously with this pest on common bean materials—from 14.13 to 15.59 [36]. Two groups were established (Table 5). For the genotypes BRS Esteio, Uirapuru, Quero-Quero, Verdão, Gralha, IAPAR 81 and Tuiuiú, a higher larval phase was observed (from 15.6 to 16.7 days) when compared with the remaining genotypes. At least 1.9 days were necessary for changing instar, which corroborates previous reports, indicating two days for switching [45].

The larval survival rate varied according to genotypes (Table 6). Two groups were found. The lesser viabilities were found for Uirapuru, ANFC 09 and Juriti: 75.36, 75.36 and 74.11%, respectively. These data are quite similar from those reported previously for rearing this larva on common bean genotypes [39].

The pupal period was short for Gralha and intermediate for the Capitão genotype (Table 5). The lowest pupal survival rate was observed for the Uirapuru genotype (25.7%). For Tangará and Eldorado, the highest values were found (84.3 and 82.9%, respectively). In previous studies, pupal variability for these two genotypes varied from 26.6 to 73.3%, respectively [39].

Lighter females were found for the genotypes Tuiuiú, Uirapuru, Quero-Quero, Gralha and Verdão (Table 7). Males feeding on Uirapuru and Gralha were lighter than IAPAR 81, ANFC 9, BRS Esteio, Tangará, Curió, Capitão, Eldorado and Juriti.

In general, pupae weight is an important Lepidoptera biological parameter since mass accumulation is related to the female reproductive performance, affecting mating, oviposition and egg survival rate [46,47]. The lesser SL pupae weight found for some genotypes may be associated with poor nutritional or antinutritional conditions. 

Females previously fed on Uirapuru, IAPAR 81 and ANFC 9 lived longer (16.8, 13.8 and 12.6 days, respectively) than females from Quero-Quero, Gralha and Tangará (7.9, 8.0 and 9.7 days, respectively). Males that fed on IAPAR 81, Tangará and ANFC 9 (14.3, 13.8 and 13.9 days, respectively) were more longevous than Capitão, BRS Esteio, Uirapuru and Quero-Quero (10.2, 10.0, 6.2 and 6.0 days, respectively) (Table 5).

The data obtained indicate that common beans genotypes may provide different nutritional quality for C. includens since adult longevity and reproductive performance are related to the larvae, such as adult nourishment [44]. The influence of the nutritional quality in these parameters was previously characterized in a study feeding *Rachiplusia nu* (Guenée 1852) (Lepidoptera: Noctuidae) with leaves of different ages [48].

Similar fecundity was observed among common bean genotypes (Table 7). In general, a similar standard of oviposition was observed, in which maximum fertility was observed in the first six days after the female’s emergence (Figure A1), except for the Campos Gerais genotype, for which females’ max oviposition was at 14 days, and for Uirapuru, for which females reached the maximum oviposition at 20 days.

### 3.5. Fertility Table

The R0 (net reproductive rate) are variable according to genotypes, varying from 106.0 to 257.1, which indicates 142% of variation among genotypes (difference between higher and lower values). Lower R0 values were obtained for Uirapuru, Capitão and Campos Gerais (Table 8). These results evidence differences in genotypes, providing suitable conditions for an increasing population since they indicate the number of new females that could be generated per mother during their lifetime. For artificial diets, the estimated R0 was 192.71 [44], indicating some genotypes may increase population even more than when reared on artificial diets. However, for most of the genotypes, the R0 values were inferior to those from the artificial diet, which may indicate lack of some essential element or presence of antinutritional substance for insect development.

Lower rm values were also obtained for the Uirapuru cv. (Table 8). In general, a higher rm suggests greater possibility for the population to succeed [49]. 

The finite rate of population increase (λ) was higher than one, suggesting population growth for all genotypes assessed. In general, the greater the λ values, greater the increment of individuals in the population [50]. 

These data suggest that despite the differences in the parameters assessed by the fertility life table, all genotypes provided conditions for increasing the SL population.

### 3.6. Total Phenols and Flavonoids Analyses

Differences in the concentration of phenolic compounds between the genotypes were found (Table 9). The highest values for phenolic compounds were obtained for Uirapuru and Juriti, differing from other genotypes. It was not possible to verify differences between genotypes regarding the concentration of flavonoids

Regarding the interactions among these chemicals’ levels with the SL biological parameters, was verified a negative correlation of phenolic compounds with survival rate of larval phase (r = −0.75 and *p* > 0.01), pre-pupa (r = −0.74 and *p* > 0.01) and egg-to-adult (r = −0.63 and *p* > 0.01). Negative correlations were also verified between flavonoids and pre-pupal (r = −0.57 and *p* = 0.04) and egg-to-adult (r = −0.52 and *p* = 0.05) survival rates.

Many factors and compounds participate in the plant’s resistance towards pests, as these can either occur constitutively or be produced in response to biological stresses [51]. 

The number of phenolic compounds can express significant interference in the insect cycle, as they function as digestive inhibitors, binding to digestive enzymes, inactivating them or producing free radicals [52,53]. Although the Uirapuru genotype was suggested as a standard of susceptibility for preference assays in the present study, based on previous field consultants’ reports, overall, biological data and the calculated fertility life table parameters suggested this genotype as the most resistant among the tested materials. 

In summary, there was no oviposition preference of the SL’s adults among the evaluated common bean genotypes, although correlation could be established with leaves’ luminosity (L*), indicating that genotypes with dark leaves are less preferred for oviposition. Soybean looper preference, biological parameters and fertility life tables were also affected by the different genotypes. Common bean genotypes with higher levels of phenolic compounds and flavonoids reduce SL survival.

The Uirapuru genotype is highlighted for its overall negative biological effect on *Chrysodeixis includens*. Thus, the Uirapuru genotype could be a candidate for plant breeding programs that seek pest-resistant materials as an alternative control method to synthetic chemical spraying, further enhancing integrated pest management programs. The genotype Tangará should not be recommended for areas with historic *Chrysodeixis includens* infestations given its overall positive effect on the pest’s biology. Evaluation of phenolic and flavonoid compounds contents in common bean genotypes could be an additional plant resistance indicator given the negative correlations with the survival rate of soybean loopers. Other secondary compounds should also be studied, as the interactions between these could prove to be key factors towards understanding plant–insect interactions and resistance patterns. The study of leaf coloration revealed that darker green leaves negatively affected C. includens oviposition, this should be studied further and possibly implemented in other plant–insect repellent studies.

## Figures and Tables

**Figure 1 insects-14-00905-f001:**
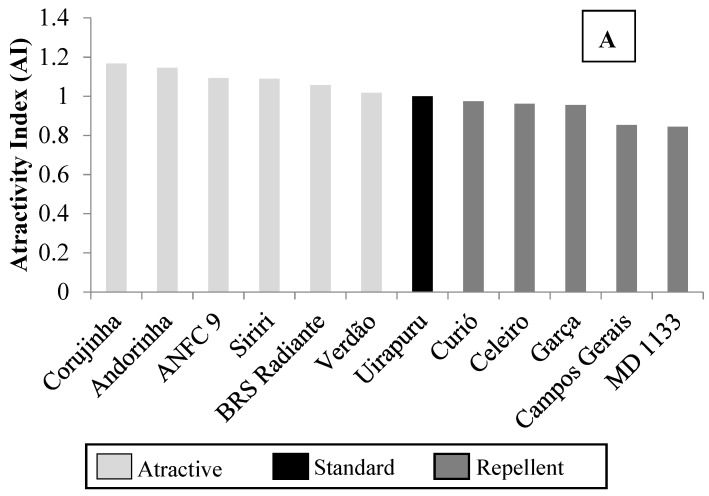
Attractivity index (AI) of *Chrysodeixis includens* (Lepidoptera: Noctuidae) larvae to common (*Phaseolus vulgaris*) bean genotypes in free-choice tests. Londrina, Paraná State, Brazil, 2019 ((**A**) = group 1; (**B**) = group 2).

**Figure 2 insects-14-00905-f002:**
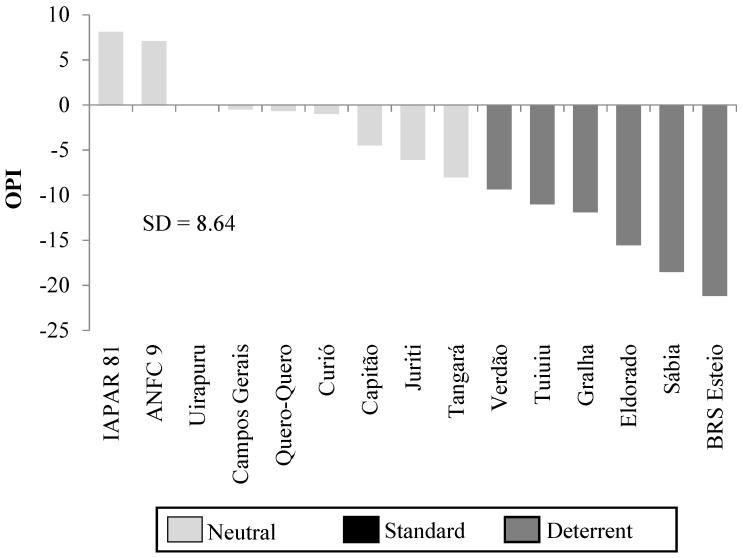
Oviposition preference index (OPI) of *Chrysodeixis includens* (Lepidoptera: Noctuidae) by common beans (*Phaseolus vulgaris*) genotypes in free-choice tests. Londrina, Paraná, Brazil. 2019 (SD = standard deviation).

**Table 1 insects-14-00905-t001:** Common bean genotypes assessed regarding resistance to *Chysodeixis includens* with varied development cycles and types. Londrina, Paraná State, Brazil, 2019.

Group	Genotype	Phenotype Characteristics
GROUP 1	BRS Radiante	Early development brindle-type bean
	MD 1133	Early development pinto-type bean
	Garça	Early development white-type bean
	Verdão	Early development colored-type bean
	Curió	Early development pinto-type bean
	Campos Gerais	Indetermined type II development pinto-type bean
	Siriri	Indetermined type II development pinto-type bean
	Corujinha	Indetermined type II development spotted-type bean
	Andorinha	Early development pinto-type bean
	ANFC 9	Indetermined type II development pinto-type bean
	Celeiro	Indetermined type II development pinto-type bean
GROUP 2	Juriti	Indetermined type II development pinto-type bean
	IAPAR 81	Indetermined type II development pinto-type bean
	Capitão	Indetermined type II development pinto-type bean
	Tuiuiú	Indetermined type II development black-type bean
	Quero-Quero	Indetermined type II development pinto-type bean
	Tangará	Indetermined type II development pinto-type bean
	Nhambu	Indetermined type II development black-type bean
	Sabiá	Indetermined type II development pinto-type bean
	Juriti	Indetermined type II development pinto-type bean
	BRS Esteio	Indetermined type II development black-type bean
	Eldorado	Indetermined type II development pinto-type bean

**Table 2 insects-14-00905-t002:** Leaf feeding (mg) by third-instar *Chrysodeixis includens* (Lepidoptera: Noctuidae) larvae in common bean (*Phaseolus vulgaris*) genotypes, Londrina, Paraná State, Brazil, 2019.

Group 1			Group 2		
Genotype	Leaf Feeding		Genotype	Leaf Feeding	
ANFC 9	0.00249	a	Gralha	0.00212	a
Celeiro	0.00196	b	IAPAR 81	0.00142	a
MD 1133	0.00139	c	Uirapuru	0.00115	b
Siriri	0.00132	c	Capitão	0.00103	b
Corujinha	0.00094	d	Tuiuiú	0.00095	b
Uirapuru	0.00078	d	Quero-Quero	0.00087	b
Garça	0.00076	d	Tangará	0.00083	b
Curió	0.00074	d	Nhambu	0.00081	b
BRS Radiante	0.00057	e	Sabiá	0.00056	b
Campos Gerais	0.00048	e	Juriti	0.00043	b
Andorinha	0.00039	e	BRS Esteio	0.00042	b
Verdão	0.00033	e	Eldorado	0.00030	b
CV (%)	2.6			4.43	

Means followed by the same lowercase letter do not differ using Scott–Knott test (*p* ≤ 0.05).

**Table 3 insects-14-00905-t003:** Average (±SD) of *Chrysodeixis includens* eggs in common bean (*Phaseolus vulgaris*) genotypes, Londrina, Paraná State, Brazil, 2019.

Genotype	Plant Structure			Total
	Unifoliate Leaf	1st Trefoil	2nd Trefoil	
Campos Gerais	39.6 ± 10.1 ^ns^	39.6 ± 16.7 ^ns^	32.8 ± 14.8 ^ns^	123.0 ± 39.6 ^ns^
Esteio	32.8 ± 17.5	28.0 ± 14.1	19.0 ± 10.1	80.80 ± 26.7
Uirapurú	29.0 ± 18.7	34.2 ± 25.4	34.6 ± 27.7	100.8 ± 73.0
ANFC-9	27.2 ± 16.0	33.4 ± 12.2	22.4 ± 14.4	85.40 ± 34.4
Sabiá	58.0 ± 25.1	49.8 ± 22.5	31.4 ± 17.2	143.2 ± 59.0
Juriti	35.8 ± 8.20	29.0 ± 9.60	31.6 ± 9.10	110.0 ± 28.1
Quero-Quero	40.6 ± 23.2	40.2 ± 17.9	35.6 ± 19.8	122.6 ± 57.6
Eldorado	35.0 ± 26.0	34.0 ± 23.5	17.8 ± 9.90	90.80 ± 58.1
Capitão	42.4 ± 13.7	46.0 ± 14.9	22.4 ± 19.2	113.6 ± 47.0
Curió	44.8 ± 16.1	38.0 ± 19.0	29.0 ± 9.30	121.8 ± 41.0
Verdão	46.2 ± 22.2	28.8 ± 21.0	22.6 ± 7.40	103.0 ± 52.1
Tangará	39.4 ± 16.9	45.2 ± 6.40	17.2 ± 7.40	105.8 ± 14.3
Gralha	42.2 ± 18.4	30.2 ± 14.0	22.4 ± 21.7	97.80 ± 40.0
IAPAR-81	40.6 ± 6.1	49.8 ± 23.1	48.4 ± 27.0	146.2 ± 58.0
Tuiuiú	42.4 ± 20.9	26.6 ± 10.3	28.2 ± 15.5	99.6 ± 44.1
CV (%)	41.12			

^ns^ Not significant by Scott–Knott test (*p* ≤ 0.05).

**Table 4 insects-14-00905-t004:** Color component values composed of brightness (L*), hue (a*) and saturation (b*) in common bean (*Phaseolus vulgaris*) genotypes’ leaves, Londrina, Paraná State, Brazil, 2019.

Genotype	L*	a*	b*
ANFC 9	34.98 ± 0.31 c	−15.58 ± 0.35 a	17.58 ± 0.51 c
Eldorado	38.06 ± 0.41 a	−17.02 ± 1.31 b	22.0 ± 0.51 a
Juriti	35.57 ± 0.24 c	−15.65 ± 0.26 a	17.49 ± 0.46 c
Uirapuru	38.08 ± 0.53 a	−17.73 ± 0.31 b	22.04 ± 0.73 a
Sabiá	35.19 ± 0.27 c	−15.44 ± 0.27 a	17.49 ± 0.51 c
Tuiuiú	35.64 ± 0.32 c	−15.63 ± 0.33 a	18.81 ± 0.92 c
Curió	34.98 ± 0.20 c	−15.84 ± 0.23 a	17.68 ± 0.36 c
Iapar 81	35.09 ± 0.49 c	−15.25 ± 0.48 a	17.50 ± 0.82 c
Campos Gerais	35.42 ± 0.28 c	−14.57 ± 0.36 a	16.18 ± 0.58 c
Tangará	37.00 ± 0.68 b	−17.10 ± 0.39 b	20.51 ± 1.03 b
Quero-Quero	37.06 ± 0.44 b	−15.58 ± 0.39 a	18.32 ± 0.76 c
Verdão	35.46 ± 0.24 c	−14.78 ± 0.34 a	16.59 ± 0.48 c
BRS Esteio	38.71 ± 0.57 a	−18.30 ± 0.55 b	22.78 ± 0.89 a
Capitão	35.06 ± 0.26 c	−17.05 ± 0.29 b	20.22 ± 0.69 b
Gralha	39.17 ± 1.81 a	−15.39 ± 0.30 a	17.46 ± 0.44 c
CV (%)	8.25	15.08	18.09

Means followed by the same lowercase letter in the column do not differ by Scott–Knott test (*p* ≤ 0.05).

**Table 5 insects-14-00905-t005:** Duration (means ± SD) in days of the development stages and adult longevity of *Chrysodeixis includens* (Lepidoptera: Noctuidae) with common bean (*Phaseolus vulgaris*) genotypes (T = 25 ± 3 °C, RU = 70 ± 10% and 14:10/L:D). Londrina, Paraná State, Brazil, 2019.

Genotypes	Eggs	Larval Instars					
		1st	2nd	3rd	4th	5th	6th
Campos Gerais	3.2 ± 0.43 e	2.0 ± 0.04 g	2.0 ± 0.09	1.9 ± 0.15	2.0 ± 0.10 b	2.2 ± 0.14 e	2.3 ± 0.55
Esteio	3.4 ± 0.48 de	2.3 ± 0.19 ab	2.0 ± 0.10	2.0 ± 0.08	2.0 ± 0.09 b	2.5 ± 0.11 bcd	2.2 ± 0.68
Uirapurú	3.7 ± 0.47 ab	2.2 ± 0.09 abcd	2.0 ± 0.10	2.0 ± 0.08	2.1 ± 0.19 a	2.8 ± 0.33 ab	3.1 ± 0.48
ANFC-9	3.5 ± 0.50 cd	2.0 ± 0.22 efg	2.0 ± 0.12	2.0 ± 0.11	2.1 ± 0.10 a	2.1 ± 0.18 f	2.4 ± 0.65
Juriti	3.5 ± 0.50 bcd	2.1 ± 0.10 bcdef	2.0 ± 0.07	2.0 ± 0.10	2.1 ± 0.16 a	2.5 ± 0.30 cd	2.6 ± 0.78
Quero-Quero	3.2 ± 0.43 e	2.1 ± 0.10 bcdefg	1.9 ± 0.10	2.1 ± 0.09	2.1 ± 0.09 a	2.8 ± 0.12 a	2.6 ± 0.69
Eldorado	3.4 ± 0.49 de	2.0 ± 0.04 fg	1.9 ± 0.08	1.9 ± 0.06	2.1 ± 0.07 a	2.6 ± 0.11 abc	2.9 ± 1.13
Capitão	3.5 ± 0.50 abcd	2.2 ± 0.15 abcd	2.0 ± 0.16	2.0 ± 0.12	1.9 ± 0.12 b	2.5 ± 0.16 cd	2.3 ± 0.76
Curió	3.4 ± 0.50 cde	2.1 ± 0.07 cdefg	2.0 ± 0.08	2.0 ± 0.09	2.0 ± 0.06 b	2.3 ± 0.16 d	2.2 ± 1.12
Verdão	3.4 ± 0.49 de	2.2 ± 0.22 bcdef	2.0 ± 0.09	2.0 ± 0.05	2.2 ± 0.08 a	2.6 ± 0.10 abc	2.5 ± 0.55
Tangará	3.6 ± 0.50 abc	2.1 ± 0.08 defg	2.0 ± 0.09	2.0 ± 0.01	2.0 ± 0.06 b	2.6 ± 0.18 abc	3.0 ± 0.93
Gralha	3.5 ± 0.50 bcd	2.2 ± 0.21 abcde	2.2 ± 0.23	2.0 ± 0.16	2.1 ± 0.14 a	2.5 ± 0.18 bcd	2.7 ± 0.57
IAPAR-81	3.7 ± 0.46 a	2.3 ± 0.17 abc	2.0 ± 0.09	1.9 ± 0.15	2.1 ± 0.12 a	2.5 ± 0.13 cde	2.7 ± 0.74
Tuiuiú	3.4 ± 0.49 de	2.3 ± 0.14 a	1.9 ± 0.14	2,1 ± 0.18	2.1 ± 0.04 a	2.8 ± 0.37 abc	2.5 ± 0.65
*p*-value	<0.01	<0.01	ns	ns	0.01	<0.01	ns
CV (%) ^1^	-	-	-	-	5.15	-	29.35
**Genotypes**	**Larva instar**	**Larval period**	**Pre-pupa**	**Pupa**	**Adults**		
	**7th**				**Pre-oviposition**	**Females**	**Males**
Campos Gerais	1.5 ± 0.5	15.4 ± 1.38 b	1.9 ± 0.15 abc	7.5 ± 0.24 a	4.0 ± 1.36	10.5 ± 5.9 abcd	11.6 ± 6.7 ab
Esteio	4.0 ± 0.0	15.6 ± 1.20 a	1.9 ± 0.11 ab	7.4 ± 0.50 a	3.9 ± 0.92	10.4 ± 5.6 abcd	10.0 ± 6.4 bcd
Uirapurú	1.3 ± 0.3	16.5 ± 0.71 a	1.8 ± 0.35 abcd	7.4 ± 0.56 a	3.6 ± 1.34	16.8 ± 5.9 a	6.2 ± 6.5 cd
ANFC-9	0.0 ± 0.0	14.5 ± 0.94 b	1.9 ± 0.09 ab	7.3 ± 0.38 a	3.3 ± 0.48	12.6 ± 5.2 a	13.9 ± 5.4 a
Juriti	0.0 ± 0.0	14.9 ± 0.71 b	1.6 ± 0.32 cde	7.3 ± 0.31 a	3.3 ± 0.73	11.4 ± 6.3 abc	13.6 ± 6.9 ab
Quero-Quero	3.0 ± 1.0	16.6 ± 1.90 a	1.8 ± 0.09 bcd	7.2 ± 0.31 a	3.4 ± 0.74	7.9 ± 5.5 d	6.5 ± 4.7 d
Eldorado	1.0 ± 0.0	14.9 ± 1.63 b	17 ± 0.25 cde	7.3 ± 0.24 a	4.0 ± 1.62	12.1 ± 6.0 ab	11.0 ± 5.6 abc
Capitão	0.0 ± 0.0	14.6 ± 0.75 b	1.7 ± 0.15 cde	7.0 ± 0.34 b	4.6 ± 2.75	11.9 ± 4.8 ab	10.2 ± 6.0 bc
Curió	3.5 ± 2.1	15.5 ± 2.53 b	1.9 ± 0.12 abc	7.5 ± 0.29 a	3.9 ± 2.20	11.4 ± 4.3 abcd	11.3 ± 6.5 ab
Verdão	2.8 ± 0.5	16.7 ± 1.60 a	1.7 ± 0.10 de	7.3 ± 0.18 a	3.9 ± 0.92	10.5 ± 6.6 abcd	10.4 ± 7.4 abc
Tangará	0.0 ± 0.0	14.8 ± 1.46 b	1.5 ± 0.12 e	7.4 ± 0.19 a	3.6 ± 0.75	9.7 ± 5.3 bcd	13.8 ± 5.5 a
Gralha	2.0 ± 1.4	16.6 ± 1.73 a	1.9 ± 0.18 abcd	6.6 ± 0.57 c	5.5 ± 3.57	8.0 ± 6.4 cd	12.4 ± 4.9 ab
IAPAR-81	2.5 ± 0.7	16.2 ± 1.36 a	2.0 ± 0.08 a	7.5 ± 0.25 a	3.3 ± 0.48	13.8 ± 4.3 a	14.3 ± 5.7 a
Tuiuiú	2.5 ± 1.5	16.6 ± 2.13 a	1.8 ± 0.13 bcd	7.5 ± 0.20 a	3.9 ± 0.81	12.0 ± 4.6 abc	10.7 ± 6.7 abc
*p*-value	ns	0.02	<0.01	<0.01	ns	0.04	<0.01
CV (%) ^1^	-	9.71	-	4.79	-	-	-

^1^ The trace (-) indicates CV absence due to nonparametric statistics: means followed by the same letter in the column do not differ by the Kruskal–Wallis test (*p* ≤ 0.05). Numeric values for the CV indicate parametric statistics. Means followed by the same letter in the column do not differ by Scott–Knot Wallis test (*p* ≤ 0.05). ns: No significance.

**Table 6 insects-14-00905-t006:** Average survival rate (±SD) of seventy *Chrysodeixis includens* (Lepidoptera: Noctuidae) in different development stages on common bean (*Phaseolus vulgaris*) genotypes. (T = 25 ± 3 °C, RU = 70 ± 10%; 14:10/L:D). Londrina, Paraná State, Brazil, 2019.

Genotypes	Egg ^1^	Larva ^2^	Pre-Pupa ^2^	Pupa ^2^	Egg-Adult ^2^
Campos Gerais	39.50 cd	93.21 ± 6.33 a	81.4 ± 10.7 a	61.4 ± 22.7 b	57.1 ± 25.6 b
BRS Esteio	51.00 abc	89.29 ± 2.91 a	82.9 ± 11.1 a	60.0 ± 20.8 b	57.1 ± 18.0 b
Uirapuru	45.60 bcd	74.11 ± 2.94 b	44.3 ± 21.5 c	25.7 ± 11.3 c	22.9 ± 9.5 c
ANFC 9	46.15 bcd	75.36 ± 5.34 b	67.1 ± 12.5 b	55.7 ± 9.8 b	52.9 ± 13.8 b
Juriti	47.57 abc	75.36 ± 4.49 b	65.7 ± 15.1 b	60.0 ± 12.9 b	55.7 ± 16.2 b
Quero-Quero	59.19 a	88.57 ± 3.62 a	80.0 ± 17.3 a	58.6 ± 15.7 b	55.7 ± 12.7 b
Eldorado	41.52 bcd	94.64 ± 2.82 a	91.4 ± 12.1 a	82.9 ± 11.1 a	82.9 ± 11.1 a
Capitão	34.72 de	89.11 ± 3.41 a	82.9 ± 11.1 a	71.4 ± 17.7 b	67.1 ± 20.6 b
Curió	49.19 abc	90.71 ± 3.57 a	82.9 ± 13.8 a	67.1 ± 13.8 b	61.4 ± 17.7 b
Verdão	53.59 ab	89.64 ± 4.79 a	84.3 ± 9.8 a	68.6 ± 13.5 b	65.7 ± 12.7 b
Tangará	48.89 abc	91.79 ± 3.3 a	90.0 ± 10.0 a	84.3 ± 12.7 a	80.0 ± 12.9 a
Gralha	25.41 e	87.32 ± 3.11 a	75.7 ± 17.2 a	55.7 ± 19.9 b	50.0 ± 21.6 b
IAPAR-81	53.57 ab	86.07 ± 3.43 a	78.6 ± 21.2 a	61.4 ± 22.7 b	58.6 ± 24.8 b
Tuiuiú	48.33 abc	93.21 ± 3.9 a	84.3 ± 12.7 a	64.3 ± 11.3 b	60.0 ± 11.5 b
*p*-value	<0.01	<0.01	<0.01	<0.01	<0.01
CV (%)	- ^3^	12.31	18.6	25.6	28.9

^1^ Means followed by the same letter in the column do not differ by SNK test. ^2^ Means followed by the same letter in the column do not differ by Scott–Knott test (*p* < 0.05%). ^3^ Variance analysis by Kruskal–Wallis method does not generate CV (%).

**Table 7 insects-14-00905-t007:** Average (±SD) of female and male weight (mg), pupa deformation (%), sex ratio, fecundity and adult deformation (%) of *Chrysodeixis includens* (Lepidoptera: Noctuidae) reared on common bean (*Phaseolus vulgaris*) genotypes. (T = 25 ± 3 °C, UR = 70 ± 10%; 14:10/L:D). Londrina, Paraná State, Brazil, 2019.

Genotypes	Female Weight ^1^	Male Weight ^1^	Pupae Deformity	Sex Ratio	Fecundity	Adult Deformation ^2^
Campos Gerais	172.9 ± 25.4 a	164.5 ± 43.2 b	3.51	0.44	325.7 ± 408.4	18.60 ab
BRS Esteio	165.9 ± 20.5 a	183.5 ± 17.9 a	6.90	0.46	512.2 ± 392.5	9.76 ab
Uirapuru	154.3 ± 30.3 b	152.7 ± 32.0 c	3.23	0.32	552.6 ± 446.9	11.11 ab
ANFC 9	178.9 ± 35.0 a	199.0 ± 23.1 a	8.51	0.51	607.4 ± 495.1	5.00 b
Juriti	173.6 ± 25.9 a	180.1 ± 26.9 a	6.52	0.48	688.3 ± 552.9	14.29 ab
Quero-Quero	153.1 ± 27.8 b	165.0 ± 24.7 b	8.93	0.43	561.1 ± 395.4	37.50 a
Eldorado	179.0 ± 20.7 a	191.0 ± 18.1 a	1.56	0.49	435.3 ± 407.3	10.34 ab
Capitão	197.7 ± 30.2 a	205.3 ± 22.6 a	6.90	0.47	30.3 ± 386.3	14.00 ab
Curió	184.3 ± 17.8 a	189.5 ± 16.5 a	5.17	0.48	487.5 ± 402.1	6.38 b
Verdão	142.2 ± 21.3 b	171.8 ± 22.2 b	5.08	0.45	512.9 ± 252.9	10.42 ab
Tangará	180.1 ± 23.2 a	190.1 ± 19.3 a	0.00	0.54	573.7 ± 646.5	13.79 ab
Gralha	135.4 ± 22.9 b	142.0 ± 26.7 c	11.54	0.53	482.0 ± 384.8	13.16 ab
IAPAR-81	190.7 ± 16.2 a	195.9 ± 15.6 a	3.64	0.48	564.9 ± 404.5	6.98 b
Tuiuiú	155.2 ± 16.1 b	173.7 ± 23.2 b	0.00	0.45	482.9 ± 314.2	2.22 b
*p*-value	<0.01	<0.01	ns	ns	ns	<0.01
CV (%)	14.13	14.13	-	-	-	0.6

^1^ Means followed by the same letter in the column do not differ by Scott–Knott test (*p* < 0.05%). ^2^ Means followed by the same letter in the column do not differ by chi-square test. ns: No significance.

**Table 8 insects-14-00905-t008:** Fertility life table of *Chrysodeixis includens* (Lepidoptera: Noctuidae) in common bean (*Phaseolus vulgaris*) genotypes. (R0 = net reproductive rate; rm = intrinsic increase rate; T = generation time; Td = doubling time; λ = finite growth rate). (T = 25 ± 3 °C, UR = 70 ± 10%, 14:10/L:D). Londrina, Paraná State, Brazil, 2019.

Genotypes	R0	rm	T	Td	λ
ANFC 9	229.2 ± 46.7 ab	0.260 ± 0.013 a	20.9 ± 0.4 a	2.7 ± 0.1 a	1.297 ± 0.016 a
Campos Gerais	106.0 ± 34.3 c	0.208 ± 0.017 b	22.6 ± 0.6 b	3.3 ± 0.3 b	1.231 ± 0.021 b
Capitão	109.4 ± 31.7 c	0.217 ± 0.016 b	21.8 ± 0.7 ab	3.2 ± 0.3 b	1.242 ± 0.02 b
Curió	182.5 ± 33.7 b	0.246 ± 0.012 b	21.2 ± 0.5 a	2.8 ± 0.1 a	1.279 ± 0.015 ab
Eldorado	172.8 ± 34.5 b	0.237 ± 0.013 b	21.8 ± 0.6 ab	2.9 ± 0.2 ab	1.267 ± 0.016 ab
BRS Esteio	179.1 ± 36.7 b	0.233 ± 0.012 b	22.4 ± 0.5 abc	3.0 ± 0.2 ab	1.262 ± 0.016 abc
Gralha	173.7 ± 43.9 b	0.230 ± 0.017 b	22.5 ± 0.7 abc	3.0 ± 0.2 ab	1.259 ± 0.021 b
IAPAR 81	211.5 ± 34.7 b	0.238 ± 0.009 b	22.5 ± 0.4 b	2.9 ± 0.1 ab	1.269 ± 0.011 ab
Juriti	251.1 ± 53.9 a	0.255 ± 0.012 ab	21.8 ± 0.4 ab	2.7 ± 0.1 a	1.290 ± 0.015 ab
Quero-quero	185.8 ± 46.3 b	0.250 ± 0.017 a	21.0 ± 0.4 a	2.8 ± 0.2 ab	1.284 ± 0.022 ab
Tangará	257.1 ± 64.8 a	0.260 ± 0.013 a	21.5 ± 0.4 a	2.7 ± 0.1 a	1.297 ± 0.017 a
Tuiuiú	167.3 ± 27.2 b	0.220 ± 0.009 b	23.3 ± 0.6 abc	3.1 ± 0.1 b	1.246 ± 0.011 b
Uirapuru	106.1 ± 38.4 c	0.159 ± 0.031 c	28.7 ± 5.5 c	4.2 ± 0.7 b	1.171 ± 0.037 c
Verdão	184.7 ± 23.5 b	0.233 ± 0.008 b	22.5 ± 0.3 abc	3.0 ± 0.1 ab	1.262 ± 0.01 bc

Means with the same letter in the column do not differ by jackknife method (*p* ≤ 0.05).

**Table 9 insects-14-00905-t009:** Average absorbance values (±SD) of total phenols and flavonoids in common bean (*Phaseolus vulgaris*) genotypes. Londrina, Paraná State, Brazil, 2019.

Genotypes	Chemical Compounds	
	Total Phenols ^1^	Flavonoids
Campos Gerais	0.366 ± 0.106 b	0.2068 ± 0.145
BRS Esteio	0.2932 ± 0.122 b	0.1594 ± 0.098
Uirapuru	0.6604 ± 0.204 a	0.3124 ± 0.081
ANFC 9	0.3176 ± 0.168 b	0.236 ± 0.107
Juriti	0.5506 ± 0.138 a	0.2232 ± 0.071
Quero-Quero	0.3046 ± 0.07 b	0.2284 ± 0.091
Eldorado	0.3382 ± 0.131 b	0.1332 ± 0.058
Capitão	0.2128 ± 0.051 b	0.1666 ± 0.074
Curió	0.3126 ± 0.044 b	0.2564 ± 0.075
Verdão	0.3114 ± 0.065 b	0.2726 ± 0.059
Tangará	0.2922 ± 0.036 b	0.2096 ± 0.067
Gralha	0.3004 ± 0.14 b	0.2074 ± 0.127
IAPAR-81	0.2636 ± 0.067 b	0.117 ± 0.028
Tuiuiú	0.4396 ± 0.109 b	0.2744 ± 0.124
*p*-value	0.01	ns
CV (%)	31.09	43.55

^1^ Means followed by the same letter in the column do not differ by Scott-Knott test (*p* < 0.05%). ns = No significance.

## Data Availability

Data will be available upon request to the correspondence author.

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
