# Peer review of "Resistance of Common Bean Genotypes to Chrysodeixis includens (Walker, 1858) (Lepidoptera: Noctuidae)"

_insects, 2023, doi:10.3390/insects14120905_

Round 1

Author Response

Dear Reviewer,

We humbly thank you for the careful reading and detailed suggestions. We hope to best answer your questions and suggestions.

1) The suggestion to better discuss the Chrysodeixis includens importance and specific impact was agreed by all authors and was addressed.

2) There is currently a table in pages 3-4 commenting what we believe to be the most important information, which is bean type and development speed. The genotypes are released to market explaining cooking speed, disease resistance/tolerance, development speed and bean type. This paper comments the lack of pest resistance/tolerance descriptions at the beginning, and the data disproves the Uirapuru genotype’s susceptible label. Recommending it as a potential breeding program candidate, due to its negative biological effects over the soybean looper, not classifying its resistance/tolerance level.

3) The paper’s main objective was to discuss the effect of different common bean genotypes on the soybean looper’s biology and physiology. Not to determine the genotype’s resistance or tolerance levels. We believe this determination would involve many more factors and needs a lot more insect behavioral and plant response data. Even though we present life table, feeding preference and leaf coloration data, these serve as potential factors towards determining genotype resistance, not concrete determination data.

4) We concur that the underlying genetic and biochemical factors could aid future breeding programs. The specific mechanisms underlying resistance that were studied in the present study were life table, feeding preference and leaf coloration data, only a few antibiosis and antixenosis indicators, we believe this is not enough to concretely categorize the genotypes as resistant/tolerant, as such we recommend further studies. Given our results, the genotypes evaluated in this study, with resistance characteristics, should be further analyzed and these specific factors determined.

5) There is a lack of common bean pest resistance studies, thus we mostly compared the results to soybean assays.

6) We agree that the crop’s pest resistance durability is a threshold with great importance, however, it was not evaluated in the given paper. Furthermore, this topic must be further developed, as resistance is not permanent and the incorrect implementation, or abuse of this control method may lead to greater problems in the future.

7) We concur that these genotypes are limited to the Brazilian market. However, the methods described, and the data collected surpass the genotype limitation, demonstrating their implementation importance during the selection of new breeds in breeding programs, and also to better determine resistance/tolerance factors in existing genotypes.

8) We completely agree with your future research directions, we believe our study contributes towards it and we hope to further enhance the available scientific knowledge.

Best regards.

Reviewer 2 Report

This paper is of merit as it looks at several parameters to assess the resistance of many genotypes of Phaseolus vulgaris to the soybean looper Chrysodeixis includens. The paper’s data analysis is sound. The paper would benefit greatly from a discussion/conclusion that summarizes the major findings with respect to the different genotypes studied.  Based on all the findings, which genotypes seem to show the most promise in exhibiting resistance to the soybean looper? What can one do with the information?

Here are some additional comments:

Line 95: “14:20 L:D” should be “14:10 L:D”

Lines 140-142: Is “Dd” the same as “DWd”?  Is “DDW” the same as “DW”?   

Lines 201-205: “1,0 mL” should be “1.0 mL”.  What is methanolic acid?  “clorite” should be “chlorite”

Lines 202 and 205, what does “de” mean?

Line 302: “16,5” should be “16.5”.  “16,7 “should be “16.7”

Here are some comments issues related to the English language.  There are some incomplete sentences: lines 12-13, 27-28 (“Highlighted the soybean looper…”), and also 374-377 is an incomplete sentence. Line 31: "greenhouses conditions" should be "greenhouse conditions".  Line 57 should be "Just leaf veins...". The meaning of line 59 is not clear.  Line 148: What is "voil"?  Line 162: the word "stadium" should be "stage"?  Line 178: It is not clear why the word "dairy" is there. Line 182: Why is the word "tax" there? Line 191: What does "under environment temperature" mean? Line 218: "follow" should be "follows". In figures 1A and 1B, "Atractive" should be "Attractive" and "Repelent" should be "Repellent".  Lines 244-245 should be "....due to calculated values exceeding..."  Line 246: “compared to standard” should be “compared to the standard”. Line 247 “inside SD” should be “inside the SD”. Line 271: "by" should be "of".  Line 287: "eggs" should be "egg".  Line 337 "sex" should be "six".  Line 338, "14th" should be "14".

Author Response

Dear Reviewer,

We humbly thank you for the acute reading and detailed suggestions.

The paper’s final considerations were corrected as suggested, highlighting the genotype with greater effect on Chrysodeixis includens biology and its potential in plant breeding programs.

We agree with the further English corrections and suggestions, all of them were performed as you recommended. Here are some that we feel should be relayed:

Lines 201-205 “methanolic acid” was corrected to methanolic extract.

Lines 12-13 & 27-28 were better defined to convey understanding

Lines 140-142 The formula’s components were correctly identified

Line 191 the sentence was corrected to “in ambient conditions”

Thank you.

Best regards.

Reviewer 3 Report

This is a potentially valuable paper for management of bean cultivars and the soybean looper. However, it suffers from systemic lack of clarity and interpretation of results.  Here are specific questions and comments.

line 82+. what was the rationale or criterion for creating the two groups of cutivars? Explain "characteristics" in Table 1.  Are all cultivars equally viable in the environmental conditions used in the bioassays (temperature, light)?  Or were they developed for specific sites and conditions, in which case these should be considered for the environmental testing conditions.

l. 122. explain what exactly was observed and counted in the statement: "After 12 hours, the number of larvae per leaf disk was recorded". Why 36 larvae in each arena with 12 leaf types?  How does this measure preference?  Why not individual larvae in choice and non-choice tests, as is often done in measuring preference?  

l. 130 mentions "randomized block design" but doesn't describe the actual factors of the design.  How were blocks defined (blocking variable)?  What variable was randomized within the blocks?

l. 159. Explain purpose of leaf coloration experiments.

l. 172. Explain "larvae (70 per plate) were individualized in Petri dishes".  

l. 259-263. This explanation of leaf hue should go into the Intro or briefly in Methods to show relevance of the measurements.

l. 305 and others throughout manuscript.  What is "larval viability"?  Is it survival?  Specify.  Same for pupae.

l. 388 and final comments.  NO overall interpretation of the results is given!! What do authors conclude regarding all the measurements across the multiple genotypes, and how do results bear on life history of the insect?  How do results relate to management of the cultivars and the soybean looper?

Appendix.  What do the arrows in the graphs indicate?

Some editing of English grammar and sentence structure is needed to clarify statements throughout the manuscript. 

For example, on line 112: "Following a randomized block design using two groups of 12 genotypes and 15 rep112 licates (arenas), totaling 360 leaf disks." is not a sentence.

Similarly, on line 13: "Highlighted the soybean looper, which feeds on leaves and pods." is not a sentence.

Author Response

Dear reviewer,

We have made the suggested corrections we believe would further enhance the paper’s understanding and value.

1) line 82+. what was the rationale or criterion for creating the two groups of cutivars? Explain "characteristics" in Table 1.  Are all cultivars equally viable in the environmental conditions used in the bioassays (temperature, light)?  Or were they developed for specific sites and conditions, in which case these should be considered for the environmental testing conditions.

  1. A) The rationale was based on facilitating the essays, and the group separation was pure randomization. This was now included in the paper. The growing conditions were equal across both groups, those being, greenhouse conditions, same soil and handling, temperature and relative humidity. Basically hoping to mimic field conditions with greater control over environmental factors. In relation to specificity of each genotype, in Brazil there are three possible common bean harvest seasons during the year, thus, the adaptation of these genotypes towards climate and regions is broad. We believe there is no need in the present study to environmentally test each genotypes adaptation to the greenhouse growing conditions.

2) 122. explain what exactly was observed and counted in the statement: "After 12 hours, the number of larvae per leaf disk was recorded". Why 36 larvae in each arena with 12 leaf types?  How does this measure preference?  Why not individual larvae in choice and non-choice tests, as is often done in measuring preference? 

  1. A) The recorded number of larvae present on each leaf disk was used to calculate the attractivity index (AI). This was made clearer in the paper. The 36 larvae were used in this study due to the shear amount of treatments, a total of 23 genotypes, if we were to cross-examine two genotypes per choice assay, it would take several days to accomplish given the great number of combinations possible, 66 combinations per replicate, or a total of 990 assays, this would most definitely be an influence on consumption amount and feeding preference given the time scale and larvae generation gap. To ward unwanted time constraints we chose to use three larvae per disk. This was further developed in the paper.

3) 130 mentions "randomized block design" but doesn't describe the actual factors of the design.  How were blocks defined (blocking variable)?  What variable was randomized within the blocks?

  1. A) The description of the block layouts was better explained as requested.

4) 159. Explain purpose of leaf coloration experiments.

  1. A) A new paragraph in the introduction was included to better explain the importance of leaf coloration and its possible use in insect resistant plant breeding programs. Also, a new line was added in the material and methods leaf coloration section to define the experiment’s purpose.

5) 172. Explain "larvae (70 per plate) were individualized in Petri dishes". 

  1. A) This paragraph was corrected and made more understandable.

6) 259-263. This explanation of leaf hue should go into the Intro or briefly in Methods to show relevance of the measurements.

  1. A) With the paragraph added in the introduction, we believe this sentence fits in the discussion as a pathway to discuss our findings.

7) 305 and others throughout manuscript.  What is "larval viability"?  Is it survival?  Specify.  Same for pupae.

  1. A) This confusing description was adjusted to “survival rate”,

8) 388 and final comments.  NO overall interpretation of the results is given!! What do authors conclude regarding all the measurements across the multiple genotypes, and how do results bear on life history of the insect?  How do results relate to management of the cultivars and the soybean looper?

  1. A) This was addressed and a new paragraph was added commenting main results and giving future study suggestions.

9) Appendix.  What do the arrows in the graphs indicate?

  1. A) The missing explanation was added.

10) Some editing of English grammar and sentence structure is needed to clarify statements throughout the manuscript.

  1. A) All these mistakes were corrected as requested, and the methods were better described.

Thank you for all the suggestions and corrections.

Round 2

Reviewer 3 Report

I was able to view the corrections and agree the manuscript can be published in its new version.

I was able to view the corrections and agree the manuscript can be published in its new version.

Author Response

Dear Reviewer,

We thank you for the valid inputs and suggestions,

We are glad you agree with the publication.

Thank you.